# Assessment of Critical Shear Stress and Threshold Velocity in Shallow Flow with Sand Particles

Reza Shahmohammadi [1], Hossein Afzalimehr [2] and Jueyi Sui [3],*

1 Department of Water Engineering, Isfahan University of Technology, Isfahan 8415683111, Iran; Reza.shahmohammadi@gmail.com
2 Department of Civil Engineering, Iran University of Science and Technology, Tehran 1684613114, Iran; hafzali@iust.ac.ir
3 School of Engineering Program, University of Northern British Columbia, Prince George, BC V2N 4Z9, Canada
* Correspondence: jueyi.sui@unbc.ca; Tel.: +1-250-960-6399

**Abstract:** In this study, the incipient motion of four groups of sand, ranging from medium to very coarse particles, was experimentally examined using an acoustic Doppler velocimeter (ADV) in different water depths under the hydraulically transitional flow condition. The transport criterion of the Kramer visual observation method was used to determine threshold conditions. Some equations for calculating threshold average and near-bed velocities were derived. Results showed that the threshold velocity was directly proportional to both sediment particle size and water depth. The vertical distributions of the Reynolds shear stress showed an increase from the bed to about 0.1 of the water's depth, after performing a damping area, then a decrease toward the water surface. By extending the linear portion of the Reynolds shear stress in the upper zone of the damping area to the bed, the critical shear stress, particle shear Reynolds number, and critical Shields parameter were calculated. Results showed that the critical Shields parameter was located under the Shields curve, showing no sediment motion. This indicates that the incipient motion of sediment particles occurred with smaller bed shear stress than that estimated using the Shields diagram in the hydraulically transitional flow region. The reason could be related to differences between the features of the present experiment and those of the experiments used in the development of the Shields diagram, including the approaches to determine and define threshold conditions, the accuracy of experimental tools to estimate critical shear stress, and sediment particle characteristics. Therefore, the change in the specifications of experiments from those on which the Shields diagram has been based led to the deviation between the estimation using the Shields diagram and that of real threshold conditions, at least in the hydraulically transitional flow region with sand particles.

**Keywords:** incipient motion; threshold condition; critical shear stress; threshold velocity; hydraulically transitional flow

## 1. Introduction

Reservoir dams for flood control, power generation, and water supply for irrigation and municipalities and industries involve huge costs. The service life of reservoirs depends on the amount of sediment delivered by rivers. On a sediment bed, if the flow velocity increases so much that the hydrodynamic forces, including drag and lift, exceed the stabilizing forces resulting from the particles' submerged weight, the sediment particle motion is intermittently and randomly initiated. The state of flow that is just sufficient to start sediment particle motion is called the threshold or critical condition [1]. The threshold of sediment particle motion plays an important role in many river engineering issues and some special problems, e.g., [2–4].

In field or laboratory experiments, the threshold condition can be determined by two methods—the bed-load extrapolation method and the visual observation method [1]. In the

bed-load extrapolation method, the critical shear stress is defined by extrapolating paired measurements of bed shear stress and bed-load transport rate to zero or low reference transport rate of sediment flux [5]. This method is sensitive to the way of extrapolation [6] and the determined reference transport value [7]. Abbott and Francis [8] classified grains movement into three different types, namely, (1) rolling at which shear stress is only a little more than the critical value, (2) grains ballistic jumps or saltation influenced by bed mean shear stress and roughness, and (3) suspended motion known by generally longer trajectories. Against suspended movement, rolling and saltation are limited to near the bed. They named the ratio of the shear velocity to critical shear velocity as the transport stage. In the transport stage, it is not expected the value is less than 1.0, but with values larger than 1.0 rolling, saltation and suspension could be detected. The visual observation method is based on monitoring the sediment particle movement. Kramer [9] defined four levels of sediment movement [5], i.e., the first stage (no sediment transport), with no movement of sediment particles; the second stage (weak sediment transport), with the movement of a small number of the smallest particles in the isolated parts of the bed; the third stage (medium sediment transport), with the motion of a large number of medium-sized particles, considering bed-surface configuration is not affected; and the fourth stage (general sediment transport), with the motion of all sizes of particles in all parts of the bed, considering it is strong enough to change the bed-surface configuration. Different definitions of threshold conditions in various investigations have led to conflicting results and have made it difficult to compare [5,10,11].

Some research studies show that the sediment motion is influenced by the near-bed turbulence, indicating the nature of hydrodynamic forces acting on the grain particles [12]. Bialik [13] applied a Lagrangian perspective to study numerically the role of the coherent structure in the incipient motion of sediment particles. He used a 3D relevant model of grains, in which a special procedure has been designed to generate coherent structures. The numerical results showed that the sweeps and outward events play a generally dominant role in the initiation of particles saltating. Dey et al. [12] attempted to quantify the turbulence characteristics of near-bed flows in threshold conditions of non-cohesive sediments. Their analysis of experimental data measured in flows over immobile and threshold condition beds showed the changes in the turbulence characteristics due to differences in bed conditions. They applied quadrant analysis of the data of velocity fluctuations and concluded that sweep events are the dominant mechanism toward sediment movement, and ejection events are prevalent at the top of the wall–shear layer. In this condition, the turbulent dissipation exceeds the turbulent generation. Nikora et al. [14] illustrated a physically based explanation of the dispersion relation, introducing two types of sand movement in the form of sand waves related to the region of small and large wavenumbers. They explained that the formation of small sand waves is a result of the individual sand particles' motion, while larger sand waves form due to the motion of smaller waves.

Threshold average and near-bed velocities are equal to the average flow velocity and near-bed (at the sediment particle level) velocity under the defined threshold conditions, respectively [1]. Numerous studies have provided equations for estimating the threshold average and near-bed velocities, corresponding to water depth and sediment particle characteristics [15–17]. However, the precise size of sediment particles and hydraulic flow regime have usually not been clearly demonstrated, and the determination of the threshold average and near-bed velocities is still a challenging issue [1], which requires more in-depth studies, especially in the presence of non-cohesive sand particles at the hydraulically transitional flow condition. On the other hand, Einstein [18], Velikanov [19], Yalin [20], and Ling [21] investigated the effect of lift force on sediment motion. However, the effect of drag force, along with that of lift force, on the threshold conditions for sediment motion should also be considered [1]. A theoretical sediment threshold condition model should consider the effect of both lift and drag forces against the particle stabilizing force resulting from the submerged particle's weight. Regarding the effect of bed-shear stress on

the threshold of sediment particle motion, all studies have only been based on laboratory measurements to yield empirical equations with different and approximate results [1].

The semi-theoretical method proposed by Shields [22] had phenomenally improved the estimation accuracy of the threshold condition [1]. Even today, it is the most commonly used method to estimate the threshold condition of non-cohesive sediments. The data points located on the Shields curve represent the threshold conditions and the regions above and below the curve represent the regions with and without sediment motion, respectively. With regard to hydrodynamic conditions, the Shields diagram is divided into three different flow regions [1], i.e., the hydraulically smooth flow with the particle shear Reynolds number less than 2, the hydraulically transitional flow with the particle shear Reynolds number between 2 and 500, and the hydraulically rough flow with the particle shear Reynolds number more than 500. In these regions, the viscous sublayer thickness is larger, almost equal to, and smaller than sediment particle diameter, respectively. The critical Shields parameter has a minimum value (about 0.032) when the particle shear Reynolds number ranges from 9 to 20 and a constant value of about 0.056 in the hydraulically rough flow region.

Despite extensive applications of the Shields diagram, many researchers have challenged its validity [5,11,23–27]. Regarding the rough turbulent flow, it is reported that the results of the Shields diagram for the incipient motion of coarse materials are not appropriate [28]. In this case, Neil [29] claimed the critical Shields parameter equal to 0.03 for the particle shear Reynolds number more than 500, while Gessler [30] obtained 0.046 for a similar condition. Unreliability of the Shields diagram makes it problematic as a recommendation for engineering applications with coarse materials, which expresses results should be divided by the number of two [28]. Some studies such as Miller et al. [23] and Yalin and Karahan [31] reported a wide range of the Shield parameter from 0.02 to 0.065. Other studies have been attempted to improve Shields' results [32–36], but the issue has remained challenging, particularly about the hydraulically transitional flow region of the Shields diagram linked to sand particles, which is related to the curved zone of the Shields diagram. Considering that most studies concentrate on the incipient motion of the coarse materials including hydraulically rough flow, it must be valuable to study the threshold condition of the sand particles in the hydraulically transitional flow regime experimentally.

Most of the fundamental studies for developing practical equations to determine the threshold average and near-bed velocities and non-dimensional critical shear stress trace back to the time when advanced experimental equipment for collecting data were not available, and the main focus was on the hydraulically rough flow. Nowadays, advancement in experimental tools provides new opportunities to improve the results of the previous experimental studies. In this study, to estimate the sand particles threshold condition in the hydraulically transitional flow accurately, the vertical distribution of velocity and Reynolds shear stress were acquired more accurately using an ADV instrument. Additionally, the exact flow discharge was measured using an electromagnetic flowmeter. Thereafter, compared to the results of previous studies, more accurate experimental results about threshold conditions have been obtained.

## 2. Theoretical Background

### 2.1. Threshold Average Velocity Equations

The following equation was developed by Goncharov [15] to calculate the threshold average velocity $U_{cr}$:

$$U_{cr} = 1.07\sqrt{\Delta g d}\log\left(8.8\frac{h}{d_{95}}\right), \tag{1}$$

where $h$ is the water depth, $d$ is the median sediment grain size, $d_{95}$ is the size for which 95% of the sediment particles are smaller, $g$ is the gravitational acceleration, $\Delta$ is the submerged relative mass density of sediment particles, and $\sqrt{\Delta g d}$ has a dimension of velocity. This equation was derived for turbulent flow around the sedimentary bed, indicating a hydraulically rough flow regime. Goncharov [15] claim that the equation does not vary

for cases that lower part of the grain particles is located inside the laminar boundary layer, which indicate equation maybe could be considered for hydraulically transitional flow regime. Neill [16] developed the following equation for estimating the threshold average velocity for coarse gravel particles for hydraulically rough flow condition:

$$U_{cr} = 1.41\sqrt{\Delta g d}\left(\frac{h}{d}\right)^{1/6}. \tag{2}$$

Based on a large amount of data in hydraulically rough flow under the threshold condition, Garde [17] proposed the following equation:

$$U_{cr} = \sqrt{\Delta g d}\left(0.5 log\frac{h}{d} + 1.63\right). \tag{3}$$

### 2.2. Threshold Near-Bed Velocity Equation

Garde [17] proposed the following equation for calculating the threshold near-bed velocity in hydraulically rough flow:

$$u_{cr} = 1.51\sqrt{\Delta g d}. \tag{4}$$

### 2.3. Shields Approach

Shields claimed that nondimensional critical bed shear stress should be a function of shear Reynolds number [28]. In the Shields diagram, the below-mentioned particle shear Reynolds number $R_*$ and critical Shields parameter $\Theta_c$ have been considered on the horizontal and vertical axis, respectively:

$$R_* = \frac{k_s\sqrt{\tau_{oc}/\rho}}{v} \stackrel{k_s=d}{\Rightarrow} R_* = \frac{d\sqrt{\tau_{oc}/\rho}}{v} \tag{5}$$

$$\Theta_c = \frac{\tau_{oc}}{\Delta \rho g d}, \tag{6}$$

where $k_s$ is the Nikuradse's equivalent roughness, which is usually assumed to be equal to the sediment median grain size, $\rho$ is the mass density of water, and $\tau_{oc}$ is the critical shear stress that, divided by $\Delta \rho g d$, is converted into the non-dimensional critical bed shear stress.

## 3. Materials and Methods

Experiments were conducted in a rectangular flume that is 15 m long, 0.9 m wide, and 0.6 m deep (Figure 1). In order to observe the movement of sediment particles, sidewalls of the flume were made of transparent glass. A slide-gate was located at the end of the flume, which allowed the water to spill over into a downstream reservoir. This reservoir was equipped with a sediment trap and was connected to the suction pipe of a centrifugal pump. The pumping system circulated the water between the flume downstream reservoir and the water tank located upstream of the flume. In order to monitor the water temperature during the experiment, a floating electrical thermometer was put in the upstream water tank.

In the upstream tank, a multi-layer grid and a secondary stilling basin were installed to dissipate water energy and reduce water flow oscillation at the flume entrance. The pumping system consisted of an electromotor pump with a discharge capacity of 50 lit/s, a piping system, an electromagnetic flowmeter, a three-phase switchboard, and a variable frequency drive. The pump discharge was controlled by adjusting the input frequency of the electromotor pump using the variable frequency drive. The electromagnetic flowmeter was installed on the outlet pipe of the pump and measured the discharge within the maximum relative percentage error of 0.5%. In addition, a liminimeter (Laboratory of The Isfahan University of Technology, Isfahan, Iran) with a measuring resolution of 0.5 mm was used to acquire the water depth. By changing the flow discharge and adjusting the end slide gate, it was possible to reach the desired flume water depth and velocity.

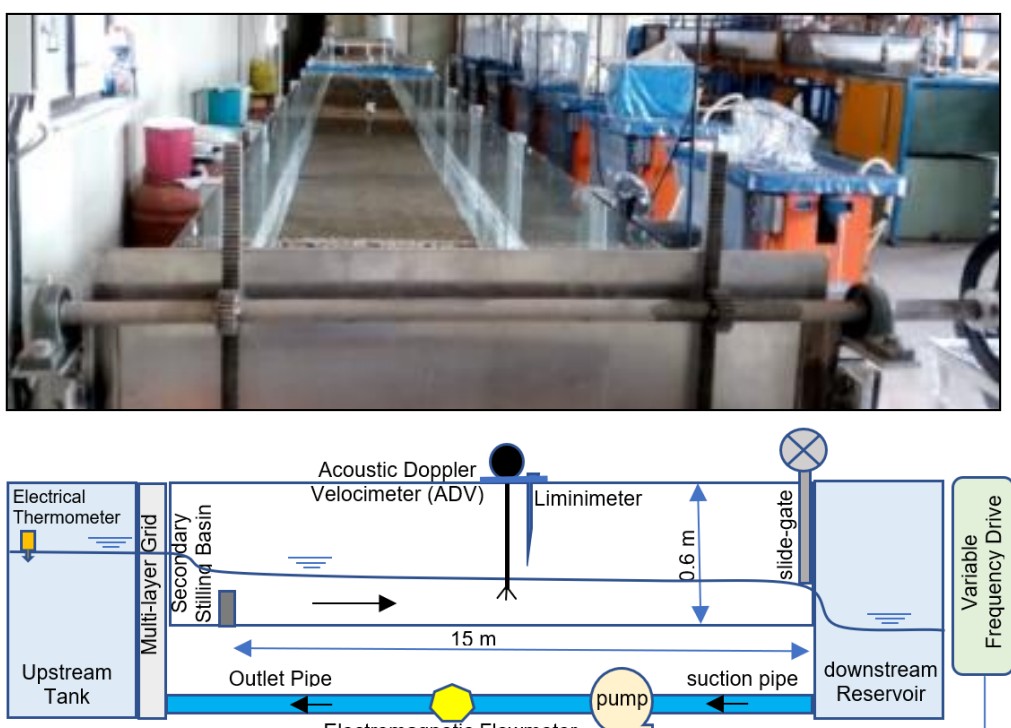

**Figure 1.** Flume used for experiments (dimensions: 15 m long, 0.9 m wide, and 0.6 m deep).

In this experimental study, an acoustic Doppler velocimeter (ADV) was used to record velocity time series data and determine velocity fluctuations. This ADV is manufactured by the Nortek Corporation with a maximum 0.5% relative percentage error and acoustic frequency of 10 MHz according to instrument instruction available at www.nortek-as.com (accessed on 1 October 2004). The ADV acquires three-dimensional velocity data including u (streamwise), v (spanwise to the left side), and w (vertical toward water surface). The negative values of w indicate a downward flow from the water surface toward the bed. The duration for data collection lasted 2 min at each point with a sampling frequency of 200 Hz, according to the latest version of the Vectrino Plus interface software (version 1.22, Nortek Corporation, Vangkroken, Norway). In this way, 24,000 data were recorded at each point along verticals for velocity measurements. The ADV was set up at each desired position using a three-dimensional moveable device. Flow velocities along all verticals along the centerline of the flume were measured. Along each vertical, approximately 20-point velocities were recorded from the sand bed to the water surface so that 50 percent of the points were in the inner layer of the velocity profile (20 percent of the water depth near the bed).

The nominal distance of the ADV transmit transducer to the focal point of sampling volume was about 50 mm. On the other hand, in order to prevent it from interference with air bubbles of the water surface, it required at least 10 mm submergence of the transmit transducer and its four receiving transducers. As a result, due to the ADV inherent limitation, the first point for velocity measurement from the water surface was at least 60 mm below the water surface. Similarly, due to the effect of the sand bed on the sampling volume, it was impossible to measure flow velocities in a zone of about 3–4 mm near the bed. Therefore, measurements of velocity along verticals were limited to a range from 3–4 mm above the sand bed to 60 mm below the water surface. It is noticeable that ADV is affected by Doppler noise and spikes caused by aliasing of the Doppler signal due to the shifting phase between the outgoing and incoming pulse [37]. The velocity data were filtered using WinADV software (version 2.024, Bureau of Reclamation, Washington, DC, USA) and aliases. Spikes were removed using the phase–space threshold despiking filter, developed by Goring and Nikora [38] and modified by Wahl [39], together with a

minimum acceptable correlation coefficient of 70 and signal-to-noise ratio of 15. On average, 18% of the data were ignored and the rest of the data were verified for analysis. Using those valid data, velocity and Reynolds shear stress profiles were analyzed and plotted by means of a program that was developed using the Excel visual basic for application (VBA) programming language.

Natural quartz sand with a mass density ($\rho_s$) of 2.65 g/cm$^3$ was used as sediment particles in this experimental study. Therefore, the relative mass density of sediment particles ($S$) and the submerged relative mass density of sediment particles ($\Delta$) were equal to 2.65 and 1.65, respectively. As shown in Table 1, after screening and grading, four groups of sediment particles numbered as I, II, III, and IV, were obtained with median grain sizes of 0.43, 0.83, 1.38, and 1.94 mm, respectively. According to the criterion for sediment size classification [40], group I is medium sand, group II is coarse sand, and groups III and IV are very coarse sand. On the other hand, the median diameter of all four material groups is less than 2 mm, indicating the range size of sand material [41]. In Table 1, $d_i$ is the size (mm), which is smaller than $i$ percent of sediment particles, and $\sigma_g$ is the geometric standard deviation of sediment particles, $\sigma_g = (d_{84}/d_{16})^{0.5}$. The values of $\sigma_g$ less than about 1.4 indicated a uniform distribution of sediment particles [42]. According to those values in Table 1, sediment particles have an acceptable uniform distribution. In Figure 2, the gradation (particle-size distribution) curves of sediments are shown in a semi-logarithmic graph. As observed, the gradation curve patterns indicated a uniform gradation.

**Table 1.** Grain size characteristics of four groups of screened and graded sediments—mm.

| Sediment | Class | $d_{16}$ | $d_{35}$ | $d_{50}$ | $d_{84}$ | $\sigma_g$ |
|----------|-------|----------|----------|----------|----------|------------|
| I | Medium Sand | 0.27 | 0.36 | 0.43 | 0.56 | 1.45 |
| II | Coarse Sand | 0.72 | 0.78 | 0.83 | 1.11 | 1.24 |
| III | Very Coarse Sand | 1.22 | 1.31 | 1.38 | 1.58 | 1.14 |
| IV | Very Coarse Sand | 1.71 | 1.84 | 1.94 | 2.28 | 1.16 |

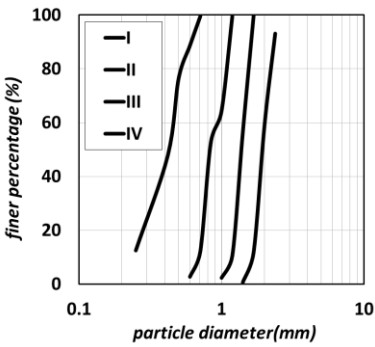

**Figure 2.** Gradation curves of sediment particles (from left to right are I, II, III, and IV, respectively).

The upstream edge of the sand bed was 8 m downstream from the flume entrance and the downstream boundary of the sand bed was 12 m downstream from the flume entrance. Namely, this 4 m long flume section was covered by sand with a thickness of 3 cm. Along this sand-bed section, the turbulent flow was fully developed, and the influence of the tailgate was negligible. As shown in Figure 3, for all sediment groups, the velocity distributions have a similar pattern in successive cross sections P1, P2, and P3, which were 9 m, 10 m, and 11 m downstream from the flume entrance, respectively. The legends in Figure 3 are explained as follows: for example, "II-H3-P3" describes a velocity profile over a sand bed with sediment group "II" and water depth of "H3" at the location of "P3". One can infer from Figure 3 that all velocity profiles at P1, P2, and P3 have similar shapes; in particular, the vertical velocity distributions at P2 and P3 are nearly the same. In this study, velocity profiles at P3 were used in the hydraulic analysis.

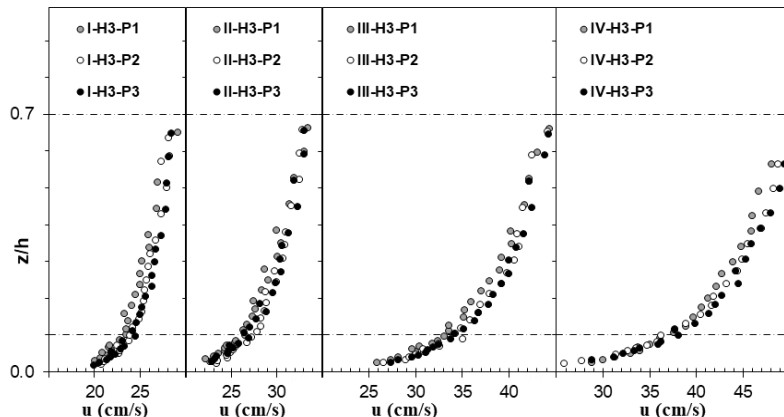

**Figure 3.** Flow velocity profiles at cross sections P1, P2, and P3 within the study reach (cm/s).

The velocity verticals were acquired using the ADV along this 4 m long sand-bed flume section. As shown in Figure 4, the upstream and downstream of the study sand-bed reach were covered by coarse particles that were not mobilized during experimental runs. The coarse-grained material upstream of the sand-bed facilitates a fully turbulent flow condition. In order to eliminate the effect of the bed roughness change on flow characteristics, the first (P1) and the last (P3) positions were located at a distance of one meter from the course materials. The position of P2 was located between P1 and P3, namely, in the middle of the study sand bed.

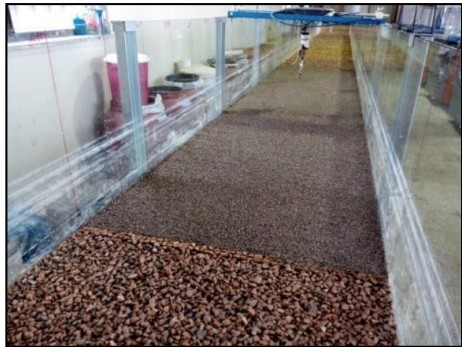

**Figure 4.** A view of the sand-bed section with coarse particles upstream and downstream.

The criterion for the incipient motion of sediment particles was determined based on the medium transport of the Kramer visual observation method that defined the movement of a large quantity of medium-size particles. To achieve the threshold conditions, firstly, the liminimeter was set up at the appropriate level. During experiments, the desired water depth is acquired by touching the pinpoint of the liminimeter on the water surface. In order to prevent sediment particles from being washed away at the beginning of the experiment run, the pump was started with a low flow rate (of about 5 lit/s), while the downstream slide gate was closed. In this way, water was gradually spilled into the flume from the tank upstream of the flume. Considering the closure of the end slide gate, the water level in the flume gradually increased. Once the water level touched the pinpoint of the liminimeter, the flume end slide gate was opened so slightly as to create a low-velocity flow for the intended water depth by creating an equilibrium state, namely, the discharge spilled into the flume equals to the flow rate out of the end slide gate. At the desired water depth, to achieve threshold conditions, the flow velocity should be increased. Thus, the pump discharge and the end slide gate opening were increased step by step, maintaining the balance between the inflow into the flume and the outflow from the flume by maintaining the intended water depth. After reaching the threshold conditions, to make sure that the motion of the sediment particles was stable, the continuous movement of

sediment particles was kept for 10–15 min. One can infer from Figure 5 that after finishing the ADV data acquisition, a number of sediment particles were eroded from the studied sand-bed section and deposited downstream reach, implying that the threshold condition was determined appropriately.

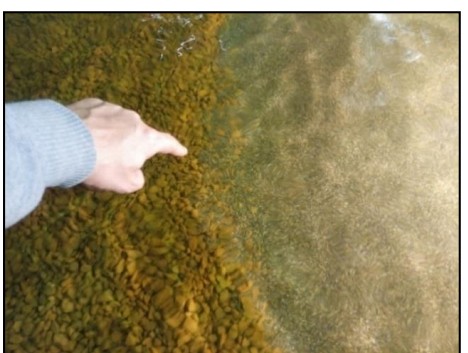

**Figure 5.** Transportation of a number of sediment particles out of the studied sand-bed reach.

## 4. Results

### 4.1. Characteristics of Flow over the Sand Bed

Table 2 shows the results and characteristics of experiments for the incipient motion of sediment particles. The experiments for each sediment group I, II, and III were conducted with three water depths of H1 = 100, H2 = 120, and H3 = 140 mm, respectively. For sediment group IV, experiments were carried out for water depths of H1 = 91, H2 = 104, and H3 = 120 mm, respectively.

**Table 2.** Results and characteristics of experiments for incipient motion of sediment.

| Experiment Name: | I (d = 0.43 mm) | | | II (d = 0.83 mm) | | | III (d = 1.38 mm) | | | IV (d = 1.94 mm) | | |
|---|---|---|---|---|---|---|---|---|---|---|---|---|
| | H1 | H2 | H3 | H1 | H2 | H3 | H1 | H2 | H3 | H1 | H2 | H3 |
| Water Depth (h)—mm | 100 | 120 | 140 | 100 | 120 | 140 | 100 | 120 | 140 | 91 | 104 | 120 |
| Relative Submergence (h/d) | 235 | 282 | 329 | 120 | 144 | 168 | 72 | 87 | 101 | 47 | 54 | 62 |
| Relative Roughness (d/h) × $10^{-3}$ | 4 | 4 | 3 | 8 | 7 | 6 | 14 | 11 | 10 | 21 | 19 | 16 |
| Aspect Ratio (B/h) | 9.0 | 7.5 | 6.4 | 9.0 | 7.5 | 6.4 | 9.0 | 7.5 | 6.4 | 9.9 | 8.7 | 7.5 |
| Water Temperature—°C | 17 | 15 | 19 | 18 | 20 | 19 | 21 | 22 | 17 | 24 | 24 | 21 |
| Discharge (Q)—L/s | 21.3 | 27.3 | 33.4 | 27.1 | 33.0 | 38.5 | 33.2 | 41.6 | 49.3 | 35.7 | 41.6 | 49.3 |
| $U_{cr}$ = Q/(Bh)—cm/s | 23.6 | 25.3 | 26.5 | 30.1 | 30.6 | 30.6 | 36.9 | 38.5 | 39.1 | 43.6 | 44.4 | 45.7 |
| Re = Uh/ν × $10^3$ | 24 | 30 | 37 | 30 | 37 | 43 | 37 | 46 | 55 | 40 | 46 | 55 |
| Fr = U/(gh)$^{0.5}$ | 0.24 | 0.21 | 0.19 | 0.31 | 0.26 | 0.22 | 0.38 | 0.33 | 0.28 | 0.49 | 0.44 | 0.39 |

### 4.2. Threshold Average Velocity

Based on experiments data, the relationship between two dimensionless parameters of $\frac{h}{d}$ and $\frac{U_{cr}}{\sqrt{\Delta g d}}$ is plotted in Figure 6, and the following equation was obtained:

$$U_{cr} = \sqrt{\Delta g d}\left(0.0024\left(\frac{h}{d}\right) + 2.34\right); \; R^2 = 0.93 \tag{7}$$

where the threshold average velocity ($U_{cr}$) is in cm/s, $d$ and $h$ are in cm, $g$ is equal to 981 cm/s$^2$, and $\Delta$ is equal to 1.65. Table 3 summarizes the values of critical average velocities determined by Equation (7) and those estimated using equations developed by other researchers. Figure 7 illustrates flow velocity distribution under threshold conditions for the four-group sediments under different water depths.

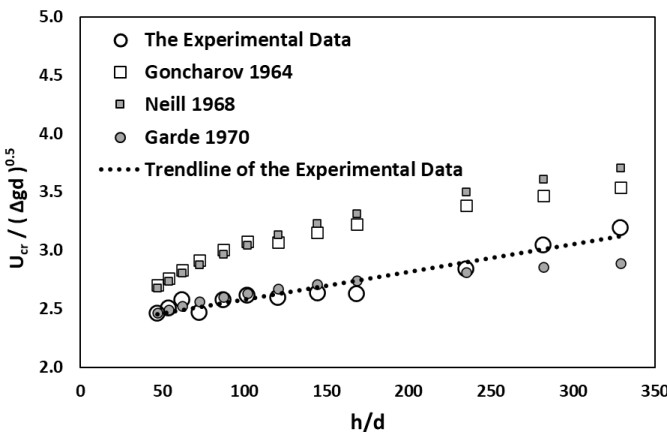

**Figure 6.** Relationship between $\frac{h}{d}$ and $\frac{U_{cr}}{\sqrt{\Delta g d}}$ in this study, compared to results of other studies.

**Table 3.** Estimation of critical average velocities obtained using Equation (7), compared to results using other equations (cm/s).

| Experiment: | | I (d = 0.43) | | | II (d = 0.83) | | | III (d = 1.38) | | | IV (d = 1.94) | | |
|---|---|---|---|---|---|---|---|---|---|---|---|---|---|
| | | H1 | H2 | H3 | H1 | H2 | H3 | H1 | H2 | H3 | H1 | H2 | H3 |
| Experimental Data | | 23.6 | 25.3 | 26.5 | 30.1 | 30.6 | 30.6 | 36.9 | 38.5 | 39.1 | 43.6 | 44.4 | 45.7 |
| Extracted Equation (Equation (7)) | | 24.1 | 25.0 | 26.0 | 30.5 | 31.2 | 31.8 | 37.6 | 38.1 | 38.6 | 43.5 | 43.7 | 44.1 |
| Goncharov | Value | 28.1 | 28.8 | 29.4 | 35.6 | 36.6 | 37.4 | 43.6 | 44.9 | 46.0 | 47.9 | 49.0 | 50.2 |
| | Difference with Equation (7) | −14% | −13% | −12% | −14% | −15% | −15% | −14% | −15% | −16% | −9% | −11% | −12% |
| Neill | Value | 29.1 | 30.0 | 30.8 | 36.3 | 37.5 | 38.4 | 43.0 | 44.4 | 45.5 | 47.5 | 48.5 | 49.7 |
| | Difference with Equation (7) | −17% | −16% | −16% | −16% | −17% | −17% | −13% | −14% | −15% | −8% | −10% | −11% |
| Garde | Value | 23.4 | 23.7 | 24.0 | 31.0 | 31.4 | 31.8 | 38.3 | 38.8 | 39.3 | 43.7 | 44.2 | 44.8 |
| | Difference with Equation (7) | 3% | 6% | 8% | −2% | −1% | 0% | −2% | −2% | −2% | −1% | −1% | −1% |

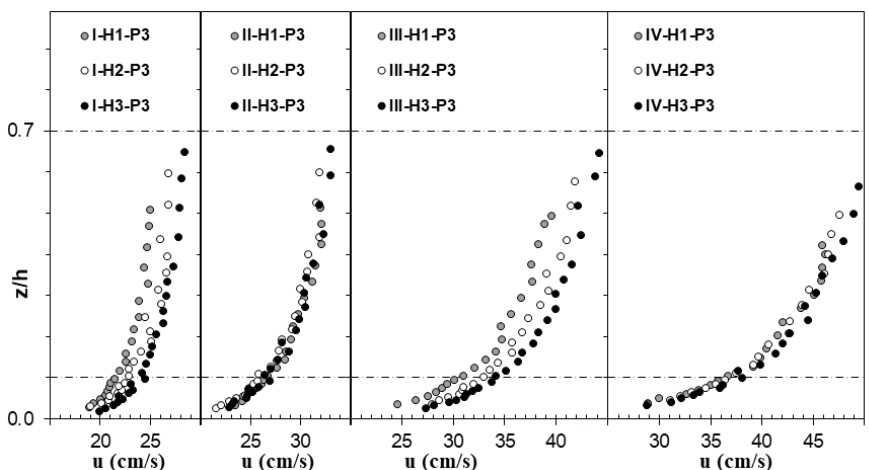

**Figure 7.** Flow velocity distribution under threshold conditions (cm/s).

### 4.3. Threshold Near-Bed Velocity

The average velocity under threshold conditions is often calculated by dividing the flow rate by the flow cross-sectional area. In reality, the flow velocity near the sand bed plays a more important role in the movement of sediment particles [1]. Since it is difficult to measure the threshold near-bed velocity precisely at the particle level, one can indirectly

obtain it by extrapolating the time-averaged streamwise velocity verticals up to the particle level [1]. By extrapolating the presented flow velocity verticals in Figure 7 toward the bed, the threshold near-bed velocity was obtained, as presented in Table 4. Considering the validity of the law of the wall in the inner layer (z/h < 0.2), the extrapolation is performed using an exponential function in the inner layer toward the adjacent of the bed.

**Table 4.** Threshold near-bed velocity (cm/s).

| Experiment | I (d = 0.43 mm) | | | II (d = 0.83 mm) | | | III (d = 1.38 mm) | | | IV (d = 1.94 mm) | | |
|---|---|---|---|---|---|---|---|---|---|---|---|---|
| | **H1** | **H2** | **H3** | **H1** | **H2** | **H3** | **H1** | **H2** | **H3** | **H1** | **H2** | **H3** |
| $u_{cr}$ (cm/s) | 18.1 | 18.2 | 19.0 | 21.5 | 20.2 | 19.6 | 22.2 | 23.5 | 24.1 | 25.3 | 24.9 | 24.8 |

Figure 8 shows the relation between the threshold near-bed velocity and sediment particle size and can be described as the following equation:

$$U_{cr} = 44.2 \, d + 16.7; \; R^2 = 0.99 \tag{8}$$

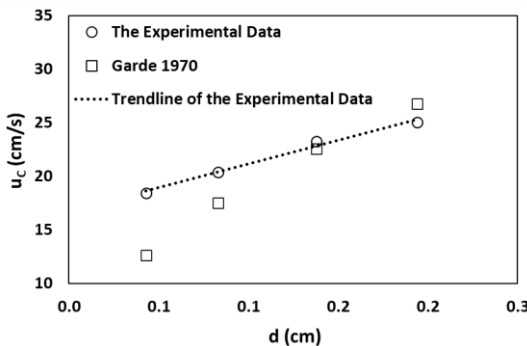

**Figure 8.** Threshold near-bed velocity vs. sediment particle size in comparison with results using equations developed by other researchers.

In Table 5, the threshold near-bed velocities, calculated by Equation (8) and equations proposed by Garde [17] (Equation (4)), are presented.

**Table 5.** Estimation of the threshold near-bed velocity using Equation (8) and those proposed by other researchers (cm/s).

| Sediment Group: | | I (d = 0.43) | II (d = 0.83) | III (d = 1.38) | IV (d = 1.94) |
|---|---|---|---|---|---|
| Averaged Experimental Data | | 18.4 | 20.4 | 23.2 | 25.0 |
| Extracted Equation (Equatiuon (8)) | | 18.6 | 20.4 | 22.8 | 25.3 |
| Garde [17] (Equatiuon (4)) | Value Difference with Equatiuon (8) | 12.6 | 17.5 | 22.6 | 26.8 |
| | | 48% | 16% | 1% | −6% |

### 4.4. Shields Diagram

To calculate the parameters of the Shields diagram (particle shear Reynolds number and critical Shields parameter), it is necessary to calculate the critical shear stress. As shown in Figure 9, the vertical distribution pattern of the Reynolds shear stress $-\overline{u'w'}$ profiles were convex, including a damping zone when the dimensionless water depth was about z/h = 0.1–0.2. The Reynolds shear stress increased from the sand bed to the water depth of z/h = 0.1 and then decreased toward the water surface. The most dominant method to estimate bed shear stress was the Reynolds shear stress distribution. This method can be used in the case of steady, uniform flow [1,28]. In this case, one should notice that there is

a near-bed damping zone, and extrapolation should be taken in the linear portion of the Reynolds shear stress distribution above the bed [1]. According to the widely accepted approach, it was extended the linear portion of the Reynolds shear stress in the upper zone of the damping zone to the bed. Then, particle shear Reynolds number and critical Shields parameter were calculated (Table 6).

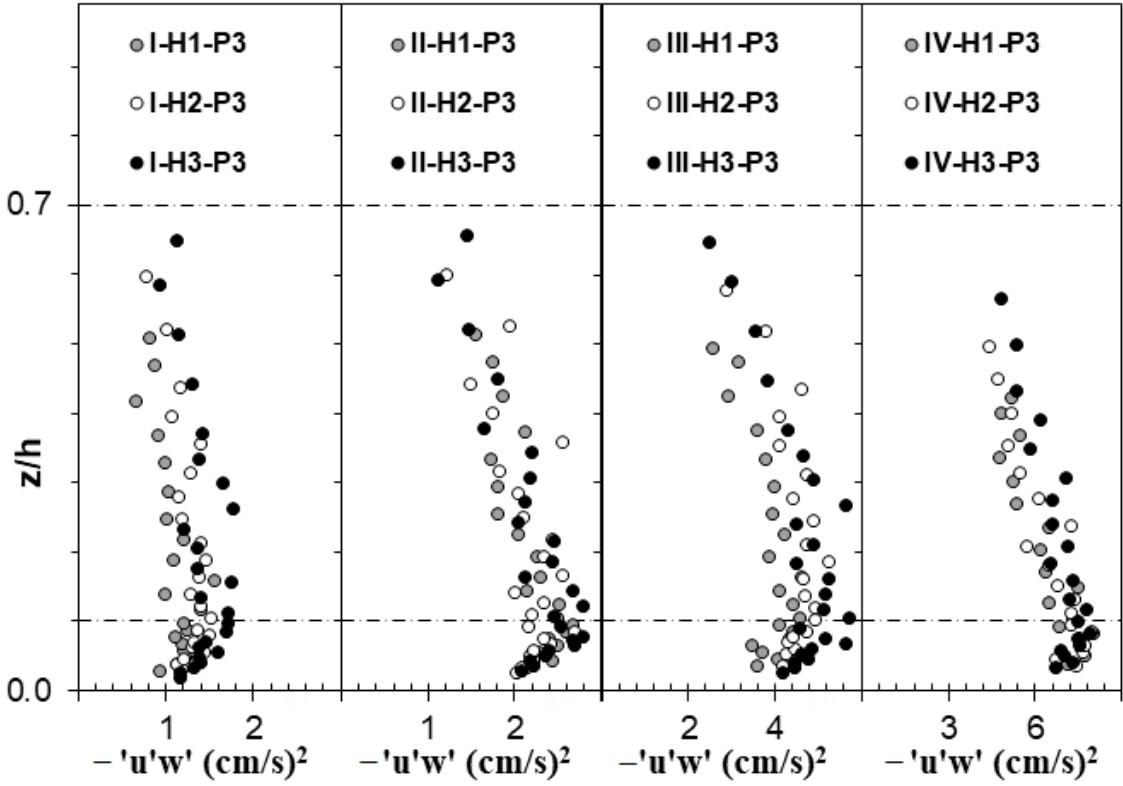

**Figure 9.** Reynolds shear stress $-\overline{u'w'}$ distributions under threshold condition $(cm/s)^2$.

**Table 6.** The critical shear stress, particle shear Reynolds number, and critical Shields parameter of the experiments.

| Experiment | I (d = 0.43 mm) | | | II (d = 0.83 mm) | | | III (d = 1.38 mm) | | | IV (d = 1.94 mm) | | |
|---|---|---|---|---|---|---|---|---|---|---|---|---|
| | H1 | H2 | H3 | H1 | H2 | H3 | H1 | H2 | H3 | H1 | H2 | H3 |
| $\tau_{oc}$ $(cm/s)^2$ | 1.5 | 1.8 | 2.2 | 2.9 | 2.9 | 3.0 | 5.7 | 6.4 | 6.9 | 8.2 | 8.4 | 9.0 |
| $R_*$ | 5.3 | 5.8 | 6.4 | 14.1 | 14.1 | 14.4 | 32.9 | 34.9 | 36.2 | 55.6 | 56.2 | 58.2 |
| $\Theta_c$ | 0.022 | 0.026 | 0.032 | 0.022 | 0.022 | 0.022 | 0.026 | 0.029 | 0.031 | 0.026 | 0.027 | 0.029 |

## 5. Discussion

### 5.1. Characteristics of Flow over the Sand Bed

Considering the values of Reynolds and Froude numbers, flow conditions in all experiments were turbulent and subcritical. The threshold average velocities for the incipient motion of sediment particles varied from 23.6 to 45.7 cm/s. The ratios of flume width $B$ to flow depth $h$, which is termed aspect ratio, varied from 6.4 to 9.9. Considering that the velocity data is acquired in the middle line of the flume with 45 cm distance from the sidewalls, and with the high enough aspect ratio, it is expected that sidewall effects and secondary currents be negligible and velocity profiles are influenced only by bed materials, including maximum water velocity at the water surface. This could be an important factor for gaining more reliable results at the threshold conditions.

### 5.2. Threshold Average Velocity

As shown in Figure 6 and Table 3, the calculated threshold average velocities using Equation (7) are in good agreement with those of experiments (presented in Table 2), in which the maximum difference between them was 2 cm/s (less than 4%). The differences between the calculated threshold average velocities using Equation (7) and equations of other researchers are less than 20%, confirming the desired performance of Equation (7). The results indicate well agreement between derived and Garde's equations.

The main reason for the difference between results using Equation (7) and those obtained by others should be related to the differences between flow regimes at the threshold condition. As already mentioned, Goncharov [15], Neill [16], and Garde [17] conducted experiments using coarse materials (gravel bed) under a rough turbulent flow regime of the threshold condition, whereas current experiments performed using sand bed under the hydraulically transitional flow regime. Additionally, it is not deniable that differences in the definition of threshold conditions and the accuracy of the velocity estimation could be other reasons for the difference. For example, in this experiment, an electromagnetic flowmeter was used to determine water discharge and velocity, which is more precise than equipment used by previous researchers.

According to Figure 7 and Equation (7), the threshold average velocity is directly proportional to the water depth. This means that by increasing the water depth, the threshold condition occurs at a higher velocity value. An increase in the sediment particle size and its submerged weight tends to increase in required hydrodynamic forces to move the particle. Thus, the threshold average velocity for the incipient motion of sediment increases with the sediment particle size.

### 5.3. Threshold Near-Bed Velocity

According to the values presented in Table 4, it can be concluded that similar to the equation proposed by Garde [17], for a specified sediment particle, changes in water depth do not significantly affect the threshold near-bed velocity. One can also infer from Tables 4 and 5 that results obtained using Equation (8) are in good agreement with the experimental data, and the difference between calculation results and those of experiments is less than 0.4 cm/s (less than 2%). According to Table 5, except for sediment group I, results using Equation (8) were in good agreement with those using equation proposed by Garde [17]. The reason for the differences in the results should be attributed to threshold average velocity because with increasing the sand particle size and increasing the particle shear Reynolds number, differences between the current experiment and Gard's equation are reduced.

### 5.4. Shields Diagram

Regarding the range of the particle shear Reynolds number, it was observed that all experimental data lay in the range of the hydraulically transitional flow. Iwagaki [33] pointed out that in the range $6.83 < R_* < 51.1$, the bed particles size is in the order of the viscous sublayer thickness. The flow is in the hydraulically transitional flow regime, and both viscous and turbulence are effective in the bed particles. Obviously, the range of the particle shear Reynolds number discussed by Iwagaki [33] is the same as this experiment condition, as presented in Table 6. As shown in the table, for a sediment group, an increase in the water depth leads to some extent larger critical shear stress. However, the effect of the water depth on critical shear stress is not noticeable, maybe due to the non-obvious differences in water depths. It is also noticed that, despite an increase in critical shear stress with the increase in sediment particle size, the variations in the critical Shields parameters for all sediment groups were not considerable, merely ranging from 0.022 to 0.032.

Figure 10 shows the relation between critical Shields parameter and relative roughness $d/h$ of the present study, compared to that presented by Buffington and Montgomery [5]. Buffington and Montgomery [5] collected experimental data using different methods to determine the incipient motion mainly through visual observation and bed-load extrap-

olation (reference transport rate). According to Figure 10, they claimed that there was an overall positive correlation between critical Shields parameter and sediment relative roughness for $d/h \geq 0.01$ (groups III and IV of the particles), and an inverse correlation for $d/h \leq 0.01$ (groups I and II of the particles).

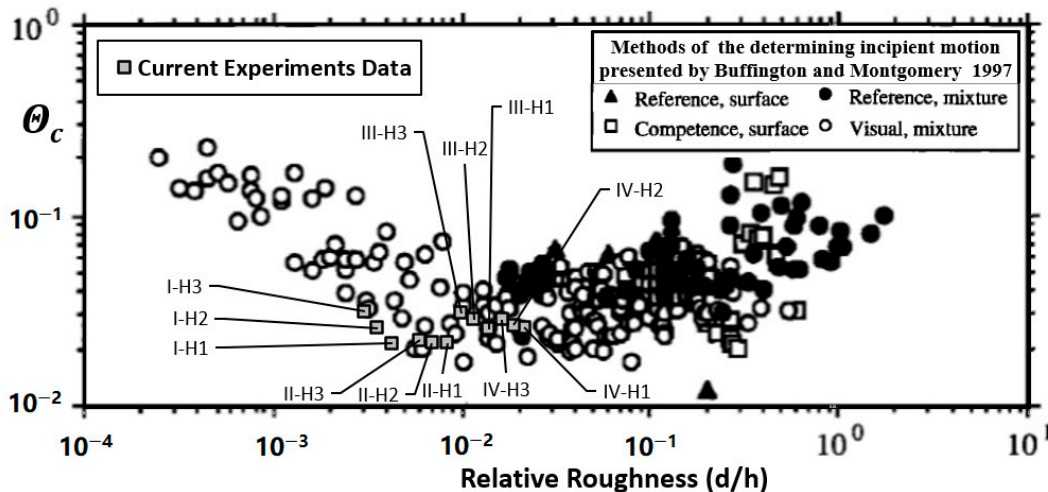

**Figure 10.** Relation between critical Shields parameter and the relative roughness of the present study, compared to those using different methods for determining the incipient motion presented by Buffington and Montgomery [5].

Experimental results are presented in the Shields diagram, as shown in Figure 11 and Table 6. It is expected that the points related to the experiments should lie on the Shields diagram curve, indicating threshold conditions. However, all data points of experiment results lay below the Shields diagram curve, where it was supposed to indicate no sediment motion. The following reasons may be attributed to this phenomenon:

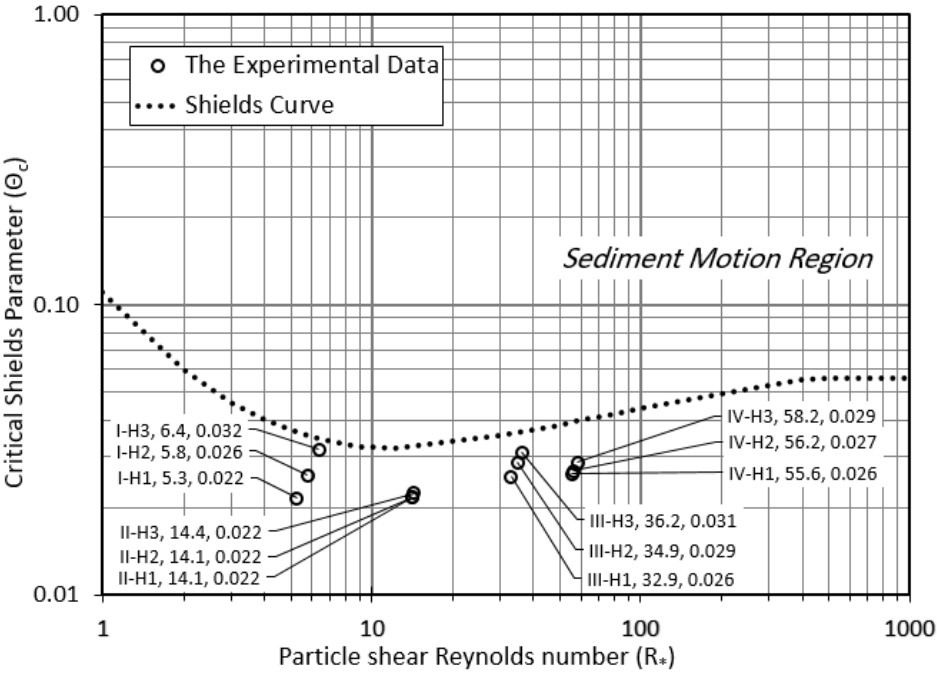

**Figure 11.** Current experimental results presented in the Shields diagram.

- Method for the definition of the threshold condition: Although Shields did not clearly explain the method, he used to determine the threshold conditions in his experiments, Kennedy [43] claimed that he probably applied Kramer's general motion criteria. Generally, there is an agreement that he used the bed-load extrapolation method to

estimate the critical shear stress [23,44,45]. In the present experiments, the medium transport criterion of the Kramer visual observation method without any bed form was used. Different methods for threshold definition could yield different values of critical shear stress. Based on a lot of different studies, Buffington and Montgomery [5] clearly showed that the bed-load extrapolation method overestimated the critical Shields parameter, compared to the visual observation method. Comparing the results of the critical Shields parameters in the present experimental study to those presented by Buffington and Montgomery [5], it clearly showed in Figure 12 that although the critical Shields parameters of the present experiments are a little smaller than those of the bed-load extrapolation method (Figure 12a), they are in agreement with those of experiments based on the visual observation method (Figure 12b). Therefore, an overestimation of the critical Shields parameters using the bed-load extrapolation method could be expected. The main reasons for the overestimation of the critical Shields parameters based on the bed-load extrapolation could be related to more probable bedform creation in the bed-load extrapolation experiments because of the method's inherent condition [46]. Bedforms influence both the bed shear stress and sediment transport by exerting more drag force and turbulence [47]. It can dissipate bed shear stress [48–50], causing significant overestimation of the bed shear stress. In addition, the negligible change of water depth owing to bedform can lead to some overestimation [47];

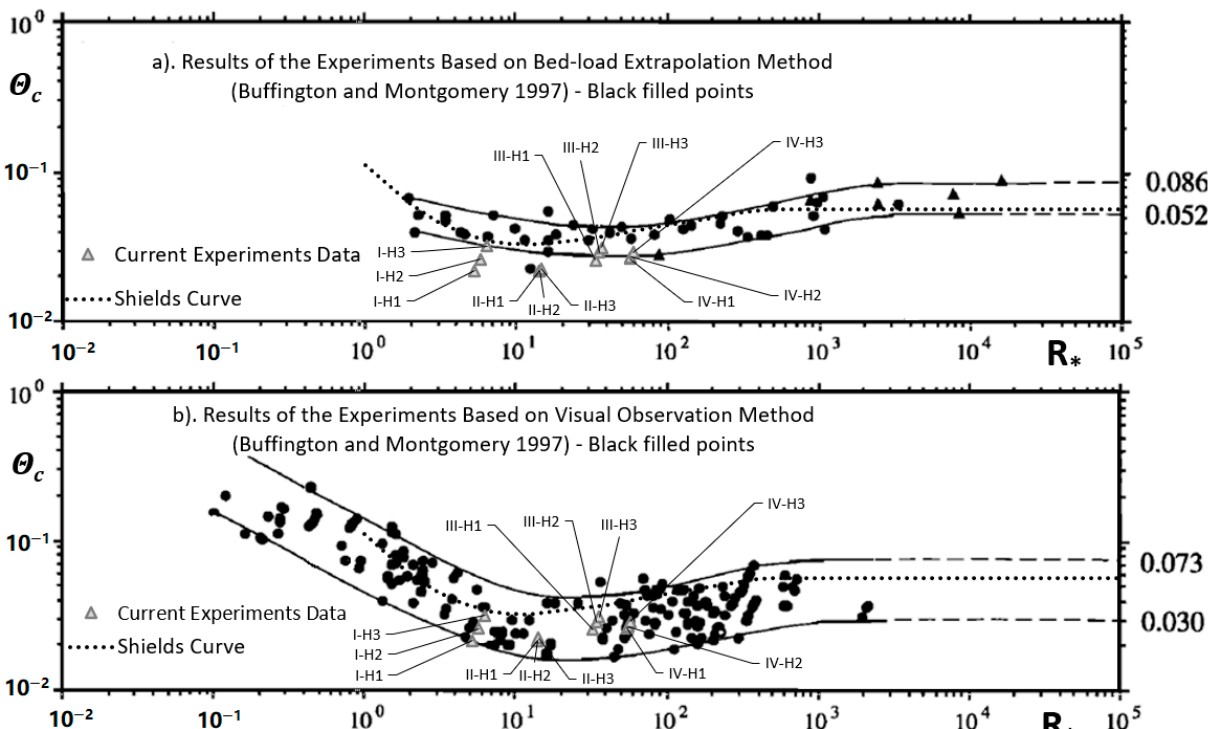

**Figure 12.** Comparison between critical Shields parameters of the present study and those of Buffington and Montgomery obtained via Bed-load Extrapolation Method (**a**) and Visual Observation Method (**b**) [5].

- Experimental measuring tools: Nowadays, the tools for acquiring vertical distributions of velocities and Reynolds shear stress in the hydraulic laboratory are more advanced, compared to those used by Shields [22] a long time ago, as also reported by Buffington and Montgomery [5]. Using an ADV in the present experimental study, it was possible to determine the critical shear stress more precisely, leading to more precise and reliable results with some expected differences from those reported by Shields [22] and Buffington and Montgomery [5];

- Sediment characteristics: The differences in sediment characteristics, such as size, shape, roundness, sorting, packing, and mass density, can affect the results of incipient motion [47,51–53]. Shields used four types of sediment including nearly uniform grains of brown coal, amber cuttings, crushed barite, and crushed granite, with the mass density ranging from 1060 to 4300 kg/m$^3$ and the median grain size ranging from 0.36 to 3.44 mm, which were sub-angular to very angular [47]. In this experimental study, naturally rounded quartz particles were used. Obviously, both crushed and angular particles cause more resistance to incipient motion due to higher friction of the particles [11], and this leads to an increase in the critical shear stress.

## 6. Conclusions

In this study, the incipient motion of four groups of sand particles has been investigated. The medium transport criterion of the Kramer visual observation method was used to determine the threshold conditions for the incipient motion. An electromagnetic flowmeter was used to measure discharge. To have reliable results, an acoustic frequency ADV was used to acquire velocity time series data. Therefore, profiles of the mean point flow velocity and Reynolds shear stress were obtained. Then, by extrapolation toward the sand bed, the near-bed velocity and critical shear stress were determined. Appropriate equations were derived to calculate the threshold average and near-bed velocities. Comparison between results using derived equations and those using equations developed by other researchers showed that there were some differences between our results and those of others. The main reason for the differences should be related to differences between bed materials and hydraulic flow regimes under the threshold condition. Additionally, the differences in the definition of threshold conditions and the accuracy of the water velocity estimation should not be ignored.

Experimental results of the current study showed that the points were located below the curve of the Shields diagram, namely, in the region for no sediment motion. Results of the present study indicate that the initiation of sediment particles occurred with bed shear stress, which is less than that estimated by the Shields diagram. This could be attributed to some differences between the current experiment and those of the Shields experiments, including approaches for determining threshold conditions, measurement accuracy of experiment tools, and sediments used in experiments. Comparison between the critical Shields parameters of the present study and those of Buffington and Montgomery [5] clearly showed that the critical Shields parameters of the present study were smaller than values related to the bed-load extrapolation method. However, these results were in satisfactory agreement with those of experiments based on the visual observation method. Results of this study support the claims of researchers who questioned the accuracy of the Shields diagram and showed that sediments start to move with smaller shear stress than that estimated by the Shields diagram.

**Author Contributions:** Conceptualization, R.S., H.A. and J.S.; methodology, R.S., H.A.; software, R.S.; validation, R.S., H.A. and J.S.; formal analysis, R.S. and H.A.; investigation, R.S., H.A. and J.S.; resources, R.S., H.A. and J.S.; data curation, R.S.; writing—original draft preparation, R.S.; writing—review and editing, R.S., H.A. and J.S.; visualization, R.S.; supervision, H.A. and J.S.; project administration, H.A. and J.S. All authors have read and agreed to the published version of the manuscript.

**Funding:** This research received no external funding.

**Institutional Review Board Statement:** Not applicable.

**Informed Consent Statement:** Not applicable.

**Data Availability Statement:** The data used in this manuscript are available by writing to the first author.

**Conflicts of Interest:** The authors declare no conflict of interest.

**Notation**

| | |
|---|---|
| $-\overline{u'w'}$ | Reynolds shear stress |
| $\tau_{oc}$ | Critical shear stress |
| U | Average velocity |
| $U_{cr}$ | Threshold average velocity (under threshold condition) |
| $u_{cr}$ | Threshold near-bed velocity (at the sediment particles level) |
| $d_{16}$ | The size of which is finer than 16% of the sediment particles |
| $d_{35}$ | The size of which is finer than 35% of the sediment particles |
| $d_{50}$ *or d* | Sediment median grain size (the size of which is finer than 50% of the particles) |
| $d_{86}$ | The size of which is finer than 86% of the sediment particles |
| $D_{95}$ | The size of which is finer than 95% of the sediment particles |
| $\sigma_g$ | Geometric standard deviation of sediment particles, $\sigma_g = (d_{84}/d_{16})^{0.5}$ |
| $g$ | Gravitational acceleration ($981\ cm^3/s$) |
| $v$ | Coefficient of water kinematic viscosity (equal to $0.01\ cm^2/s$) |
| $\rho$ | Mass density of water (equal to $1.0\ g/cm^3$) |
| $\rho_s$ | Mass density of sediment particles (equal to $2.65\ g/cm^3$) |
| $S$ | Relative mass density of sediment particles ($S = \rho_s/\rho$, is equal to 2.65) |
| $\Delta$ | Submerged relative mass density of sediment particles ($\Delta = S - 1 = 1.65$ for quarts) |
| $k_s$ | Nikuradse's equivalent roughness |
| $R_*$ | Particle shear Reynolds number |
| $\Theta_c$ | Critical Shields parameter (non-dimensional critical shear stress) |
| ADV | Acoustic Doppler velocimeter |
| $h$ | Water depth |
| $h/d$ | Relative submergence of sediment particles |
| $d/h$ | Relative roughness of sediment particles |
| $B$ | Flume width (is equal to 90 cm) |
| $B/h$ | Aspect ratio (the ratio between the flow width to the water depth) |
| $Q$ | Flow rate (water discharge measured by electromagnetic flowmeter) |
| $Re$ | Reynolds number |
| $Fr$ | Froude number |

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
