# Peer review of "Assessment of Critical Shear Stress and Threshold Velocity in Shallow Flow with Sand Particles"

_water, doi:10.3390/w13070994_

Round 1

Reviewer 1 Report

Completing the Shield's diagram is one of the basic and important tasks faced by an open channel hydraulics. In this sense, the work fits in with global trends and the presented results are worth publishing. However, it requires supplementation, in particular, the description of the research to date is very limited. I realize that it is difficult to include all the works on this subject, but a few classic ones should be there.

I see lack of two papers one that should be presented in method section:

Despiking acoustic Doppler velocimeter data

DG Goring, VI Nikora

Journal of hydraulic engineering 128 (1), 117-126

As it is interesting how Authors solved the problem presented in this paper. And the second one which is related to the movement of sand waves:

Statistical sand wave dynamics in one-directional water flows

VI Nikora, AN Sukhodolov, PM Rowinski

Journal of Fluid Mechanics 351, 17-39

This is also very problematic in calculation of condition of initial motion of particles.

I cannot also agree with the authors that there are only two methods of determination of threshold conditions. The authors completely omitted the numerical experiments but the problem has been considered by many researchers from Eulerian or Lagrangian perspectives. See for example:

Numerical study of near-bed turbulence structures influence on the initiation of saltating grains movement

RJ Bialik

Journal of Hydrology and Hydromechanics 61 (3), 202-207

In this context more info about relationship between turbulence and initiation of sand movement is required to be added to the paper.

It would also nice to see at least short info in relation to findings of Abbott and Francis (1977) who reported that the initial velocity of particles should be 2 times shear velocity. Please see:

Saltation and suspension trajectories of solid grains in a water stream.

E. Abbott, J. D. Francis; Published 1977; Geology; Philosophical Transactions of the Royal Society of London. Series A, Mathematical and Physical Sciences.

This is the authors duty to write taking into their work account state of the art papers.

Moreover, I have few minor suggestions:

In figures 3, 7, 9 the point 0.0 should be higher – due to the scale used there or 0.7 should be 0.8. Please check this.

In table 6 0c in tau should be smaller (down) index.

Figures 10-12 should be bigger for the better readability.

Author Response

Please see attached response

Reviewer 2 Report

The present work presents beneficial findings that can significantly improve the examination of particles' critical condition. The paper by Shahmodammadi et al. seems to be very interesting and fits the scope of the Water. The paper's structure is logically sound, although the work requires the improvement of some elements to be published.

General comments:

The roughness height (ks) plays a crucial role in sediment movement and critical state. this is supported by the fact that ks are also included in the Shields curve's theoretical derivation: the location of the curve also changes as a function of the roughness height. (see e.g., Fig 2-22. a, in Sediment Engineering, Chapter 2 – Marcelo Garcia). Please, mention this in detail and conduct your examinations and conclusions accordingly:

In your calculations, you use ks = d assumption. Justify this! Please, illustrate the sensitivity of the result on the applied roughness height calculation procedure (1 - 3 * d90) (chapter 3.4).

In the introduction, you briefly explain the main characteristics of hydraulic conditions. (smooth or rough). But what is the relevance of this in evaluating your laboratory test results? Some measuring points fall to a smooth region, others to a transitional region. However, Garde’s relations (Eq. 3 and 6) were developed to a rough region (you could also describe the other equations for which hydraulic region it was developed). Is it possible that the hydraulic condition influences the validity of the relationships? Please discuss the importance and role of hydraulic conditions (I recommend e.g., Török et al.: The Shear Reynolds Number-Based Classification Method of the Nonuniform Bed Load Transport).

Please, describe the used method to estimate the bed shear stress in more detail. Can it be used in hydraulically smooth, transitional and rough conditions?

Figure 10 and 12 is difficult to read and interpret. Furthermore, the legend is incomplete.

I believe that the work is a valuable study and undoubtedly worthy of publication.

Author Response

Please see attached response

Reviewer 3 Report

The authors present experimental results from a flume study pertaining to the incipient motion of sand particles. 

The focus of this work significantly lacks novelty: everything from the experimental study, experimental matrix, criteria and method to assess threshold of motion, experimental results and discussion points, have long been published, a number of decades ago (eg see the review study from Buffington and Montgomery, the authors cite too).

Unfortunately, I do not think that this study can be considered for publication, until the authors clearly demonstrate throughout the paper, sufficient novelty for their work.

Author Response

Please see attached response

Reviewer 4 Report

The submitted paper shows experimental results on the incipient motion of four groups of sediments, ranging from medium to very coarse grains. Advanced techniques based on acoustic frequency ADV were used to acquire current velocities. The results were compared to other experimental equations to determine the critical average and near-bed velocities. A discussion about the disagreement with the Shields diagram is proposed. The paper is clearly organized and presented. The experimental setup is correctly described. The results are quite well shown and discussed. I think that this paper can offer some new experimental data which can be useful for expanding knowledge on the incipient movement of sediments. I have some suggestions for the authors.

Introduction

I think that the introduction should not include equations. I suggest to report a synthesis of the main methods of calculating the critical velocities underlining the main characteristics of each method. The equations can be reported in a second section focused on some theoretical backgrounds.

Lines 58-60. The sentence requires some references.

Line 63. The symbols should be specified. These symbols are defined several lines below. Also the sentence itself can be moved.

Paragraph 90-112. It would be better to describe the reasons why the validity of the well-known Shields diagram is questioned rather than commenting on it. In this sense, the novelties of the present study can be better explained.

Section 2

I suggest to add a schematic picture of the flume.

Figure 3. Please add the title of the x-axis.

Results and discussion

The result section has to be separated from the discussion. In particular, the discussion in lines 276-279 can be expanded and better proposed. The same for lines 318-321.

I ask the authors if it is possible to add a table showing the percentage differences between the experimental results and those deduced by the other authors instead of inserting them in the text (see lines 270-273, 315-316). This can make the results more readable.

Figures 7 and 9. Please add the title of the x-axis.

Line 300. Is the extrapolation done with a linear approach? This aspect should be better specified.

Lines 333-335. A similar consideration to the previous one. I ask if the regression has been done on all values or if instead two different slopes have been considered for z / h <0 and z / h> 0.1.

Equation 10. Is the symbol ucr instead of Ucr?

Line 340. “an in increase”, perhaps “in” should be removed.

Figures 10 and 12 should be enlarged a little to be clearer.

Author Response

Please see attached response

Round 2

Reviewer 1 Report

I appreciated the changes done by the authors. The paper has been significantly improved and can be accepted for publication.

Reviewer 3 Report

The authors put some effort in working this manuscript - but not a focused effort, nor sufficient. This reviewer feels the additions are superficially connected to the current study (which is fundamentally lacking novelty) and in the backward direction (eg see additions on mean flow velocity threshold criteria), rather than moving the topical research forward. 

Specifically, the authors have made some additions on flow velocity and near bed surface flow velocity and how this may affect the transport of bed surface material (incipient motion). The literature discussed is still fairly outdated (averaged velocity criteria are not truly given any research consideration other than applied engineering practice), as is the focus on discussing the now for two decades covered, issue of averaged Shield's shear stresses. Any recent additions (Nikora, Dey ect) in the introduction/literature review, are not discussed further in essence or truly linked to this study's results.

Modern averaged concepts also require advanced instrumentation and/or methods (eg see Nikora's studies, also cited by the authors). The modern trend for analysis of incipient motion criteria is using instantaneous shear stresses (see work from Schmeeckle), or flow impulses (see work from Diplas and Valyrakis).

If the authors are limited by their available equipment, my suggestion would be to conduct studies on irregular channel configurations or geometries, that are not well studied. But even then, their work cannot be fundamental (unless generalisation of the results is achieved), and should be submitted to applied engineering journals, as a technical or case study than research work.

Unfortunately, there is nothing new or innovative in this work, to render it worth publishing in a Q1 or Q2 journal. This study, even though overall ok presented, still has no novel concept, tool or methodology of assessment that render it publishable in Water MDPI. The authors have a better chance in submitting this work to a Q3 or Q4 journal.